# Challenges and Opportunities of Biocoagulant/Bioflocculant Application for Drinking Water and Wastewater Treatment and Its Potential for Sludge Recovery

**DOI:** 10.3390/ijerph17249312

**Published:** 2020-12-12

**Authors:** Setyo Budi Kurniawan, Siti Rozaimah Sheikh Abdullah, Muhammad Fauzul Imron, Nor Sakinah Mohd Said, Nur ‘Izzati Ismail, Hassimi Abu Hasan, Ahmad Razi Othman, Ipung Fitri Purwanti

**Affiliations:** 1Department of Chemical and Process Engineering, Faculty of Engineering and Built Environment, Universiti Kebangsaan Malaysia, UKM Bangi 43600, Selangor, Malaysia; setyobudi.kurniawan@gmail.com (S.B.K.); rozaimah@ukm.edu.my (S.R.S.A.); norsakinahsaid95@gmail.com (N.S.M.S.); ezaty_ismail@yahoo.com (N.I.I.); hassimi@ukm.edu.my (H.A.H.); ahmadrazi@ukm.edu.my (A.R.O.); 2Study Program of Environmental Engineering, Department of Biology, Faculty of Science and Technology, Universitas Airlangga, Kampus C UNAIR, Jalan Mulyorejo, Surabaya 60115, Indonesia; 3Research Centre for Sustainable Process Technology (CESPRO), Faculty of Engineering and Built Environment, Universiti Kebangsaan Malaysia, UKM Bangi 43600, Selangor, Malaysia; 4Department of Environmental Engineering, Faculty of Civil, Planning, and Geo Engineering, Institut Teknologi Sepuluh Nopember, Kampus ITS Sukolilo, Surabaya 60111, Indonesia; purwanti@enviro.its.ac.id

**Keywords:** alum, coagulation, environment, flocculation, green technology, natural coagulant

## Abstract

The utilization of metal-based conventional coagulants/flocculants to remove suspended solids from drinking water and wastewater is currently leading to new concerns. Alarming issues related to the prolonged effects on human health and further pollution to aquatic environments from the generated nonbiodegradable sludge are becoming trending topics. The utilization of biocoagulants/bioflocculants does not produce chemical residue in the effluent and creates nonharmful, biodegradable sludge. The conventional coagulation–flocculation processes in drinking water and wastewater treatment, including the health and environmental issues related to the utilization of metal-based coagulants/flocculants during the processes, are discussed in this paper. As a counterpoint, the development of biocoagulants/bioflocculants for drinking water and wastewater treatment is intensively reviewed. The characterization, origin, potential sources, and application of this green technology are critically reviewed. This review paper also provides a thorough discussion on the challenges and opportunities regarding the further utilization and application of biocoagulants/bioflocculants in water and wastewater treatment, including the importance of the selection of raw materials, the simplification of extraction processes, the application to different water and wastewater characteristics, the scaling up of this technology to a real industrial scale, and also the potential for sludge recovery by utilizing biocoagulants/bioflocculants in water/wastewater treatment.

## 1. Introduction

Water is part of our life and a basic necessity for humans. It is one of the main life supports for humans. Humans need water for use as a source of body fluids and for several activities, such as bathing, washing, and using latrines. Some of these activities later cause the generation of wastewater. Treatment processing is needed to maintain the stability and continuity of the water supply. Particularly, drinking water and wastewater treatment are an important part of the water cycle in human life. 

Several treatment technologies are used to process raw water sources into drinking water and transform wastewater into treated effluent before it is discharged to water bodies, and these treatments include conventional and advanced technologies [1,2]. Most of the treatment processes, whether for water or wastewater, cannot be separated from coagulation and flocculation stages, as part of the treatment processes. Coagulation and flocculation are parts of a water treatment system that have the main function of separating suspended particles in water to produce clear and suspension-free effluent [3]. The step of the coagulation–flocculation process is normally in the primary treatment of a water or wastewater treatment system [4,5].

The processes of coagulation and flocculation require the addition of compounds known as coagulants and flocculants [6]. The main types of coagulants and flocculants used in the treatment of drinking water and wastewater are divalent positively charged chemical compounds. Negatively charged polymers are also largely used in water treatment, notably as high molecular weight flocculants [7]. The chemical compounds commonly used as coagulants/flocculants include iron salts (FeCl_3_ or Fe_2_(SO_4_)_3_) [8], aluminum salts (Al_2_(SO_4_)_3_) [8,9], hydrated lime [8], magnesium carbonate [8], and polymers (aluminum chlorohydrate, polyaluminum chloride (PAC), polyaluminum sulfate chloride, and polyferric sulfate) [10]. Some of the mentioned compounds have been shown to be effective in reducing suspended solid concentrations in water.

The application of these compounds is not necessarily free from impacts [11]. Several environmental problems due to the chronic toxicity of coagulants/flocculants are currently being discussed, specifically for environmental observers worldwide [12,13,14]. In-depth analysis has been conducted in relation to the impact that can be caused by the use of chemical compounds as coagulants and flocculants [15,16,17,18]. The environmental impacts include increasing the corrosion rate of metallic utilities [19], changing the pH, limiting root elongation, and inhibiting seed germination [20,21]. Water and wastewater treatment involving conventional coagulants/flocculants also generates excessive chemical sludge in addition to the suspended solids to be removed; thus, the handling of chemical sludge becomes another issue to resolve [12]. Aside from these impacts on the environment, concerns related to human health arise. Metallic-based coagulants/flocculants are nondegradable or nonbiodegradable, and their residuals in drinking water can induce a direct impact on human health when consumed and can be accumulated in body cells [14,22,23]. The residuals of chemical coagulants/flocculants, when used in wastewater treatment, in treated effluent discharged to the environment may be trapped in food chains [22,23]. Some indications regarding the impacts of chemical coagulants/flocculants on human health, including central nervous system failure, dementia, Alzheimer’s disease, and severe trembling, have been reported [24,25,26,27].

Biocoagulants/bioflocculants can be an alternative solution to minimize the environmental pollution and health risks caused by the use of chemical coagulants/flocculants [14,18]. Biocoagulants and bioflocculants come from living things or their parts and are totally organic and biodegradable; therefore, they are environmentally friendly and have minimal impacts on human health [28]. Research related to biocoagulants and bioflocculants has undergone many stages until their application to treatment processing units [29,30]. Some biocoagulants and bioflocculants obtained from various sources have already been analyzed and been proven efficient for application to treatment processes as a substitution for the currently widely used chemical coagulants and flocculants [30,31,32,33,34]. 

Recent articles on coagulation–flocculation are mostly standalone, do not integrate the results of the utilization of biocoagulants/bioflocculants, and have not considered the potential of resultant sludge recovery at the end of water and wastewater treatment. Compilation of the scattered findings will be useful in providing cohesive information related to the treatment of drinking water and wastewater by using biocoagulants/bioflocculants. Side-to-side comparison of the utilization of chemical coagulants/flocculants and biocoagulants/bioflocculants is also limited, especially in terms of operating dosages and overall costs. This paper presents a summary of the various sources of biocoagulants/bioflocculants. An in-depth discussion regarding the performance of biocoagulants as an alternative in water and wastewater treatment is provided. Efficiency and cost comparisons of the use of biocoagulants and chemical coagulants are juxtaposed. Current research related to the potential health risks of chemical coagulants and flocculants is also described as a basis for readers’ assessment regarding their side and long-term effects. Furthermore, this paper provides a full and in-depth description of the current research, application limits, potential to recover nontoxic sludge for other purposes (such as use as a soil conditioner, fertilizer, or feedstock), and future challenges in the use of biocoagulants/bioflocculants in drinking water and wastewater treatment.

## 2. Conventional Process Using Coagulants and Flocculants 

### 2.1. Fundamentals of the Coagulation–Flocculation Process

Coagulation is a chemical process in water treatment that requires the involvement of particle charge neutralization [35]. It is a process of coagulant addition to a solution to neutralize negatively charged particles. A coagulant is a compound with positive charges (mostly divalent) that can interact with suspended particles inside the solution and create a neutral form of combined compounds [36,37]. The coagulation processes are usually followed by flocculation and sedimentation stages in water treatment facilities, with the phases visualized in Figure 1. A specific mixing speed, intensity, and time are needed to accelerate the rate of particle collision [5,38]. The collision of flocs will create an agglomeration of particles that have a high settling velocity [39]. Several mechanisms, including charge neutralization, sweep coagulation, bridging, and patch flocculation, can occur during the formation of flocs [40,41] (Figure 2).

### 2.2. Factors Affecting the Coagulation–Flocculation Process

Several factors affect the coagulation and flocculation processes in water and wastewater treatment, including the coagulants/flocculants used, the mixing processes, and the characteristics of the water to be treated (Figure 3). The type of coagulants/flocculants will remarkably affect the performance of the coagulation and flocculation processes due to the occurrence of different mechanisms [3]; meanwhile, the dosage of coagulants/flocculants needs to be optimized before the application [5,42]. An optimum dosage of coagulants/flocculants can be obtained by plotting the measured turbidity (or any other pollutant parameter) versus the applied dosage (Figure 1). 

Mixing is crucial in coagulation and flocculation processes. The basis of mixing in the processes comprises fast mixing to promote the interaction of coagulants/flocculants with suspended particles and the formation of microflocs [38] and slow mixing to promote the aggregation of microflocs and formation of large flocs. An excessively low mixing speed and considerably short mixing time can decrease the rate of floc formation, whereas an excessively fast mixing speed and considerably long mixing time can promote floc breakdown, causing low settling efficiency [3]. 

Chemical coagulation and flocculation are processes that are highly dependent on pH changes because the latter has the ability to determine the species of polymeric polymer that will form when metal-based coagulants/flocculants dissolve in water [30,43]. The specific operational pH needs to be analyzed for each coagulant type. A change in temperature also has a substantial effect on the aggregation process [44]. The initial concentration of suspended particles influences the performance of coagulants/flocculants. Coagulants/flocculants typically work well in high-turbidity water, whereas their performance is reduced in low-turbidity water [42,45]. A coagulant/flocculant aid might be required in the process of treating extremely high- or low-turbidity water [5]. The characteristics of the suspended particles in water also affect the performance of coagulation and flocculation processes. This phenomenon is highly related to the zeta potential characteristics. 

### 2.3. Types of Coagulants and Flocculants and Their Main Applications

In terms of water and wastewater treatment, coagulants and flocculants are categorized into two types, mainly inorganic and organic. The classification of coagulant and flocculant types is illustrated in Figure 4. The aluminum-based coagulants include aluminum chloride, aluminum sulfate, sodium aluminate, and PAC [46]. The iron-based coagulants include ferric chloride, ferric chloride sulfate, ferric sulfate, and ferrous sulfate [47]. The other inorganic coagulant compounds include magnesium carbonate and lime hydrate [47]. Organic coagulants and flocculants (known as biocoagulants/bioflocculants) are usually water-soluble polymers (polyelectrolytes) originated from various natural macromolecule compounds, including polyamines, polydiallyldimethyl ammonium chloride, dimethylamine, and polyacrylamides (PAMs) [48]. For plant-based coagulants/flocculants, the functional groups of hydroxyls, carboxylic acids, and amines are considered the main ingredients that promote the functions as biocoagulants/bioflocculants [49,50,51]. The microorganism-based category mostly consists of bacteria and their extracellular polymeric substance, which can function as biocoagulants/bioflocculants [52,53]. The processes of coagulation and flocculation remove approximately 60–70% of natural organic matter [54]; thus, subsequent processes, such as filtration and oxidation, will have a lower load [4,11,55]. 

### 2.4. Criteria for Effective Coagulants/Flocculants

The effectiveness of coagulants is determined by the formation of multicharged polynuclear molecules with adsorption capability after hydrolysis. The higher the charges formed by coagulants are, the better the coagulation process will be [3,46]. The effectiveness of flocculants is determined by their ability to facilitate the formation of flocs. The larger the flocs formed are, the better the flocculation process will be [41,56]. The produced ratio of sludge to the addition of compounds is calculated as the yield and serves as the criterion for the effectiveness of coagulants/flocculants. The lower the ratio of the produced sludge/used compound is, the more efficient the coagulants/flocculants will be [57,58].

The effectiveness of aluminum and iron coagulants arises principally from their ability to form multicharged polynuclear complexes with enhanced adsorption characteristics. The nature of the complexes formed may be controlled by the pH of the system. When metal coagulants are added to water, metal ions (Al and Fe) hydrolyze rapidly, forming a series of metal hydrolysis species [3,4]. Rapid and slow mixing, the pH, and the coagulant dosage determine which hydrolysis species is effective for treatment.

For flocculants, cationic polyelectrolytes are usually favorable because most water and wastewater impurities have negative charges [28,59]. In addition, being water soluble and having a high molecular weight (MW) with a low fraction of charged monomer units are the most prominent characteristics of flocculants [60]. A high MW of flocculants is needed to form interparticle bridging among particles during processes, while linear-form polymers exhibit better bridging capability than branched form polymers (ramified structure) [7]. The low fraction of charged monomer units can maintain the surface charge of formed particles, preventing a reverse charge that could decrease the flocculation efficiency [59].

## 3. Environmental and Health Impacts of Chemical Coagulants/Flocculants

### 3.1. Toxicity and Health Risk Potential of Conventional Coagulants and Flocculants

Concerns regarding the toxicity of conventional coagulants/flocculants used in our everyday lives are emerging. These concerns are arising due to the health risks and environmental pollution potential of chemical compounds used as our daily coagulants/flocculants. Aluminum-based coagulants show a considerable increase in toxicity potential in the concentration range of 100–200 mg/L on the basis of the Microtox assay [18]. A high-MW inorganic polymer shows even more toxic potential in a lower concentration range of 400 µg/L–60 mg/L. Synthetic polymer (PAM) also poses potential carcinogenicity while also showing a very low biodegradability [61]. Not only the chemical compounds but also the sediment after treatment by using high-MW polymer-based coagulants tends to be more toxic compared with aluminum-based coagulants [18]. The excessive use of aluminum-based coagulants has been proven to affect the pH stability of water and tends to induce a great pH decrease [62]. Maintaining the operational pH in the range of 6–8 during coagulation and flocculation processes is important to prevent the release of a potentially toxic soluble form of aluminum- and iron-based coagulants. 

Several factors affect the toxicity of coagulants and flocculants. As mentioned before, pH is one of the factors that might increase the potential toxicity of chemical aluminum- and iron-based coagulants/flocculants. At an acidic pH, a positive correlation exists between aluminum-based coagulants and the inhibition of the root elongation and seed germination of plants [20]. The interaction of coagulants/flocculants with a coagulant aid can also cause an increase in their toxicity. Potential toxicity escalation has been found under the condition of the interaction of aluminum- and iron-based coagulants with a high-MW polymer used as a coagulant aid [16]. The pollutant content in a solution can cause increasing effluent sediment toxicity due to the chemical interaction between coagulants/flocculants and pollutants (mostly occurring in treating heavy metal-containing solutions). The use of a high-MW polymer can cause genotoxicity as shown by direct DNA damage and have a lethal effect on microorganisms [63]. Traces of nondegradable metals used as coagulants/flocculants can accumulate inside body cells when drinking water is consumed, or metal ion residuals can end up in food chains [22,23]. Recent findings have indicated that an excessive amount of aluminum in the human body (especially the brain) has a high correlation with neurotoxic diseases [64], Alzheimer’s disease [65], and autism [66]. However, Gillete-Guyonnet et al. [67] and Graves et al. [68] previously stated that there was no significant correlation between the metals (aluminum and silica) used for drinking water treatment and the development of Alzheimer’s disease. By contrast, several researchers have stated that the low correlation of Al with Alzheimer’s disease is due to the reason that only a very low concentration of aluminum can enter the brain and that excess amounts of aluminum will be excreted from the body [14,69,70]. In the end, the controversy surrounding alum’s neurotoxicity may become a further cause for concern and precautions regarding its potential harms. 

### 3.2. Environmental Pollution Caused by Chemical Coagulants/Flocculants

The amount of produced sludge (especially from alum) from coagulation and flocculation processes in water and wastewater treatment is large. In the UK, more than 270,000 tons of aluminum- and iron-based coagulants were used for water and wastewater treatment purposes in 2014 [71]. From that utilization, 182,000 tons of dried sludge was produced from water and wastewater treatment [72] that utilized chemical coagulants in the processes. This number is projected to be more than four times higher in 2020. On the basis of a water content of 92–99% for sludge [71], the handling of treatment sludge containing chemical coagulants would reach up to 72.8 million tons of slurry sludge per annum in the UK alone. Such an amount of sludge is dumped into regular landfills, disregarding the toxic effects of the used coagulant. The high possibility of metals leaching into the soil and groundwater are currently raised as the potential environmental pollution that might be created through this disposal pathway in the long term. 

The major issues related to the abundant amount of water treatment sludge are the disposal into regular landfills and the utilization of mature landfill soil for agricultural purposes [11,55,73,74,75]. Barakwan et al. [11] and Mortula et al. [23] discussed the possible leaching of aluminum in the form of aluminum ions or inorganic aluminum during the process of landfill dumping under a pH of 6. Fouad et al. [75] reported a high concentration of not only aluminum but also iron, manganese, and chromium from water treatment sludge, which could possibly leach in a low-pH condition. The disposal of aluminum sludge directly into water bodies, as stated by Barakwan et al. [24], may cause changing the stability in the aquatic environment, such as a decreased pH, the bioaccumulation and biomagnification of aluminum, and the further potential release of toxic metals. 

Aluminum, iron, and other metals are nonbiodegradable [76]; thus, sludge that contains these compounds will also be considered nonbiodegradable. An abundant amount of aluminum in water bodies exerts negative effects on fish and invertebrates by disturbing their osmoregulatory functions [13]. At a low concentration, below 3200 µg/L [77], aluminum seems to be safe for the environment; above the mentioned concentration, aluminum induces acute toxicity to aquatic species, such as *Danio rerio*, *Pimephales promelas*, rotifers, and freshwater snails [13]. 

Iron-based coagulants seem to have a lower sludge yield compared with aluminum-based coagulants. Despite their effectivity, the former are relatively more corrosive to piping materials [47]. The utilization of these compounds creates a lowering condition regarding the pH of the effluent and produces acidic sludge that contains iron metal. Considerable research has stated that iron is needed by organisms as a micronutrient [78,79,80,81,82,83], but an excess amount of this compound in bioavailable forms can be detrimental to the ecosystem. Despite reducing the environment’s pH condition, iron can also affect some bacterial enzymatic reactions [84] and thus can inhibit bacterial growth [85]. A substantial amount of iron inhibits plant growth. Iron toxicity mostly occurs due to the increasing amount of bioavailable iron in soil, increasing the iron uptake by plants. The excessive uptake of iron exerts a negative effect on plants, including the bronzing and stippling of leaves caused by the release of enzymes to control free radicals, causing epidermis damage on roots, reducing overall plant growth and decreasing the survival rate [86,87,88,89,90].

## 4. Characterization of Biocoagulants and Bioflocculants

### 4.1. Origin of Biocoagulants and Bioflocculants

During ancient times, people were not well exposed to proper water treatment. For normal household usage, water was just boiled and filtered to acquire consumable water. This scenario is still present in certain regions with limited access to proper water sources and technologies. As time evolved, people found a method to clarify unclean water by adding some plant powder, which is termed biocoagulant, to turbid water to settle down the dirt. In the 19th century, metal coagulants were introduced and started to gain global attention. China was the first country to use alum for wastewater treatment [91]. Chemical coagulants were continuously improved afterward to achieve the highest efficiency and suitability with the greatest operating conditions; hence, enhanced coagulants were produced, and their relevance developed over time. 

### 4.2. Chemical Characteristics

Biocoagulants can be extracted from plants, animals, or microorganisms [92,93]. The important characteristics of these sources that enable them to be used as biocoagulants are the contents of polysaccharides [94], protein polymers [92], and some functional groups [91], such as hydroxyl and carboxyl groups. Polysaccharides, protein, and some functional groups promote the mechanisms of adsorption, polymer bridging, and charge neutralization (Section 4.3). Several major compounds that could perform as biocoagulants/bioflocculants are summarized in Table 1. 

### 4.3. Working Mechanism of Natural Coagulants/Flocculants

The mechanisms of natural coagulation are mainly adsorption, charge neutralization, polymer bridging, precipitative coagulation, and electrostatic patching. The first three are the main mechanisms of biocoagulation, as described below.

Adsorption: Natural polymers provide a free surface to adsorb colloid particles and form larger particles that are easier to settle down [60,94,110].Polymer bridging: Colloid particles will attach to a part of a long-chain polymer, while the other free part of the chain will form a loop and a tail. The molecules will continue to form a larger molecule when the free tail attaches with another free colloid, increasing the particle size. The correct dosage of coagulants to provide a free surface for the process is important [92,110].Charge neutralization: Colloid particles are normally negative in charge and cannot form a larger particle because they repel one another. Thus, the addition of cationic biocoagulants will produce carboxylate and H^+^ ions to neutralize the suspension near to zero zeta potential and make the formation of a large floc possible. A low dosage of coagulants will be needed for the treatment if they have a high charge density [92,93].

Natural coagulants produce a five times lower volume of sludge compared with inorganic salts. Dorea [111] stated that this condition occurs because alum requires as many as three molecules of water hydration to fulfil its covalent bond, thus resulting in an increment in sludge volume. The sludge produced in biocoagulation is biodegradable, with high nutritional value, and it is safe and suitable for land usage (biofertilizer) [94,111,112].

In addition to being a clarification agent, biocoagulants have also been reported to have antimicrobial and heavy metal removal properties, which are effective in high-turbidity water [50,113]. Choy et al. [91] mentioned that aside from starch, phytochemicals, such as tannins and alkaloids, help in antimicrobial activities. On the contrary, natural coagulants will increase the organic matter concentration in the water, thus leading to undesired microbial activities because the antimicrobial efficiency of biocoagulants is normally low. Organic matter will also affect the color, odor, and taste of water [48,92]. Accordingly, Gunaratna et al. [113] suggested removing the content in natural coagulants through simple purification/filtration.

### 4.4. Ethical Utilization and Toxicity

In coagulation treatment, traces of coagulants may remain in treated water [92]. Hence, the usage of natural coagulants is safe, and no serious problems regarding pipe corrosion will occur due to their noncorrosive properties [91]. The application of alum to water treatment has been reported to lead to health problems, such as Alzheimer’s disease [113]. Thus, the substitution of chemical coagulants with green coagulants, which are safer, eco-friendly, and low-cost, is recommended. Natural coagulants have effectiveness comparable with that of chemical coagulants for treating wastewater but have not been successfully commercialized yet due to the lack of scientific proof of their working mechanism and efficiency [50]. 

## 5. Advantages of the Utilization of Biocoagulants/Bioflocculants

The utilization of biocoagulants/bioflocculants in drinking water and wastewater treatment has many advantages, as compiled in Figure 5. The six major advantages that can be achieved when using biocoagulants/bioflocculants are that they are an environmentally friendly technology, exhibit reliable performance, result in waste reduction/local resource utilization, are applicable in remote areas, reduce sludge production, and allow potential by-product utilization as soil conditioner/fertilizer. 

### 5.1. Environmentally Friendly Technology

The utilization of biocoagulants/bioflocculants in drinking water and wastewater treatment has been proven to be an environmentally friendly technology [43,114,115,116,117] compared with the use of conventional chemical coagulants/flocculants. This technology produces a minimum amount of harmful by-products for the environment [118]. All the involved materials in this technology will be degraded naturally by the environment. Compared with the use of conventional coagulants/flocculants, the utilization of biocoagulants/bioflocculants produces sludge with higher biodegradability, which is less harmful for aquatic biota and can be further processed biologically [56,119,120]. The utilization of biocoagulants/bioflocculants seems to be old, but it is currently gaining considerable attention as the cleanest environmentally friendly technology for coagulation and flocculation processes. 

### 5.2. Reliable Performance

This technology is not only environmentally friendly but also has reliable performance in terms of pollutant removal. The utilization of *Azadirachta indica* as biocoagulants can remove up to 97.91% of microalgae from wastewater, compared with the 91.62% removal achieved by using conventional alum [121]. A chitosan compound extracted from mushrooms achieves a removal of turbidity from palm oil mill effluent wastewater up to 99% at a low dosage of 10 mg/L, whereas 1,200 mg/L of alum can only achieve 93% turbidity removal [122]. The utilization of biocoagulants/bioflocculants can also reduce the number of Gram-positive and negative bacteria by up to 85.61% and 77.63%, respectively [123]. A study conducted by Megersa et al. [124] showed the potential of biocoagulants from indigenous plant species in treating domestic wastewater, with the highest performance of 99% turbidity removal and 98% microbial and effluent removal, which meets the quality standards for wastewater discharge. Such evidence indicates that the application of biocoagulants/bioflocculants in treating drinking water and wastewater is probable, especially in terms of turbidity removal. 

### 5.3. Waste Reduction/Local Resource Utilization

For the production of biocoagulants/bioflocculants, it is firmly suggested to utilize local resources to prevent future scarcity in raw materials. This option will benefit the local community economy by optimizing the potential of local resources [125]. Local resources are normally abundant in amount and usually have no potential utilization discovered yet. Current research is also implementing the utilization of waste, including food, agricultural, and industrial waste, in exploring the potential of biocoagulants/bioflocculants [105,126,127,128,129]. Such research is beneficial for waste reduction and finding the potential of waste to be used as biocoagulants/bioflocculants. These advantages are clearly proven, but further research on this topic is needed. 

### 5.4. Remote Area Application

The utilization of local resources creates the advantage of the application of biocoagulants/bioflocculants in remote areas. The distribution of chemicals usually hardly reaches remote areas, while the amount of local resources is normally high. The application of biocoagulants/bioflocculants in remote areas has been conducted by many researchers in terms of providing clean water for the rural communities in Bangladesh [45], Kenya [130], Southern Africa [131], Malaysia [132], India [133], Indonesia [134], and other countries. Despite the application, a challenge remains in local resource optimization, and it will be discussed further in Section 8. 

### 5.5. Sludge Generation Reduction

The utilization of conventional coagulants, especially alum, produces an enormous amount of nonbiodegradable sludge. The amount of generated sludge from drinking water and wastewater treatment had reached up to 270,000 tons of dried sludge per year in the UK in 2014 [71]. The sludge normally ends up in regular landfills because no regulation that strictly restricts handling aluminum and iron in sludge currently exists. The utilization of biocoagulants/bioflocculants has been proven to reduce up to 30% of the produced sludge from treatment processes compared with the use of alum [104,135,136]. The potential for sludge reduction will benefit all sectors in the processes, especially for the sludge-handling section. 

### 5.6. Potential of Produced Sludge Utilization

The utilization of biocoagulants/bioflocculants produces highly biodegradable sludge; therefore, sludge utilization is also possible. The further treatment of sludge through anaerobic digestion is applicable and produces valuable gas as a by-product [33]. Biocoagulants/bioflocculants have low toxic effects on the microorganisms in anaerobic digestion and will thus not interrupt their performance [58]. The current trend also indicates the potential of the utilization of generated sludge from wastewater treatment as a soil conditioner/fertilizer. The sludge from wastewater treatment in agricultural sectors, such as aquaculture, the palm oil industry, sago production, and coffee manufacturing, normally consists of high organic contents and nutrients [102,137,138,139,140]. The organic content in sludge can be further composted and used as a soil conditioner, while the available nutrients will act as a soil fertilizer. The recovered solids from urban wastewater treatment plants can consist of up to 15% phosphorus [141], which may be used in the agricultural field. In addition to the previous statement, Kominko et al. [142] reported the possibility of synthetizing an organo-mineral fertilizer from domestic wastewater. The utilization of the produced sludge will not only reduce the potential harm to the environment but could also benefit the agricultural sector [143,144]. Several sludge utilization studies are highlighted in Table 2. 

## 6. Application of Biocoagulants/Bioflocculants to Drinking Water and Wastewater Treatment 

The utilization of biocoagulants/bioflocculants shows reliable performance in treating drinking water and wastewater. Most of the parameters of pollutants in drinking water and wastewater can be removed via the utilization of biocoagulants/bioflocculants. Those parameters include the total suspended solids (TSSs), biological oxygen demand (BOD), chemical oxygen demand (COD), color, and nutrients. A summary of the performance of biocoagulants/bioflocculants in removing pollutants in drinking water and wastewater is presented in Table 3. 

To summarize the compilation of data in Table 3, the performance of biocoagulants/bioflocculants in removing pollutants (e.g., TSSs, COD, BOD, algae, and color) is undeniably great compared with that of conventional metal-based coagulants/flocculants. Biocoagulants/bioflocculants can achieve similar or even higher pollutant removal efficiency than the conventional flocculants. Most of the countries involved in the research into biocoagulants/bioflocculants are tropical and developing countries. This phenomenon can be ascribed to the abundance and diversity of potential resources (whether from waste or by-products), especially plants and crustaceans, to be utilized as biocoagulants/bioflocculants due to the tropical climate [28,134,147,148].

Plant-based biocoagulants/bioflocculants are still being specialized in this research topic. Most research has already implemented the utilization of local resources (native plants) or isolation from the indigenous environment (for microorganisms). However, research that utilizes waste or by-products to seek their potential as biocoagulants/bioflocculants is still limited [105,127,128]; further study on this particular theme could be a future direction. Most of the animal-based biocoagulants/bioflocculants come from crustacean studies [127,149] because the composition of the chitosan of crustaceans is beneficial for coagulation/flocculation. Additional study on another phylum might be interesting to provide alternative technologies. Extensive studies on fungus- and alga-based biocoagulants/bioflocculants will contribute to this topic, considering that research from these sources is currently still scarce [28,150,151,152]. 

## 7. Comparative Evaluation of Chemical Coagulants vs. Biocoagulants

### 7.1. Public Acceptance

With regard to the currently emerging issues of the utilization of metal-based coagulants/flocculants, biocoagulants/bioflocculants provide an alternative technology as a solution to overcome the potential health and environmental risks from the utilization of metal-based coagulants/flocculants. Biocoagulants/bioflocculants are totally organic materials, thus not inducing toxic effects on the environment [117]. On this basis, the utilization of biocoagulants/bioflocculants will work well in terms of public acceptance [160]. The advantages of optimizing local resources, utilizing waste or biomass/unused by-products, producing sludge minimally harmful to the environment, and not threatening humans’ health might be the public considerations for replacing the application of chemical coagulants in conventional coagulation and flocculation technology.

### 7.2. Availability and Handling of Materials

Biocoagulants/bioflocculants are suggested to be obtained from local resources, and the abundance of raw materials is considered sufficient for further utilization [134]. Raw materials for biocoagulants/bioflocculants are suggested to be sourced from waste or biomass; hence, the development of this technology might help in reducing the amount of waste or biomass [105,126,127,128,129]. The extraction processes for biocoagulants/bioflocculants are considered complex and undeveloped yet, thus making the availability of these ready-to-use compounds limited. On the contrary, the mass production of conventional coagulants and flocculants is already well developed. Therefore, ready-to-use conventional coagulants and flocculants will be easier to acquire compared with the natural ones. 

In terms of material handling, both options seem to be equal. A large number of coagulants/flocculants are usually required due to the purity of the compounds [71,119]. These chemicals need certain protocols for storage and handling to maintain their efficiency for further utilization [25,46]. By contrast, the production of biocoagulants/bioflocculants is relatively complex and inadequately maintained. In practice, the application of conventional coagulants/flocculants is already well-known and developed, and many people have specialization in this particular skill because it requires certain protocols to be followed. For the utilization of biocoagulants/bioflocculants, the development of this technology is currently mostly conducted at the research stage and laboratory scale [111,130,134]; thus, the practical applications and the number of skilled people in this subject are still relatively low. That is, the application of biocoagulants/bioflocculants requires minimum special skills and has low health and environmental potential hazards. However, further intensive research needs to be pursued to seek abundant natural coagulants, which might be waste and do not compete with foods, and develop cost-effective methods for extracting biocoagulants/bioflocculants from natural resources for bulk production and easy handling.

### 7.3. Generation and Handling of Sludge

The produced sludge from drinking water treatment typically has the characteristics of high suspended solids, COD, BOD, nutrients, minerals, and contents of some metals [24,104,128,161]. For wastewater treatment, it depends on the type of wastewater but mostly has the characteristics of high suspended solids, BOD, COD, nutrients, and contents of some minerals [119,144,162]. The contents of aluminum and iron are somewhat increasing due to the utilization of chemical coagulants and flocculants, but these parameters are normally not considered mandatory characteristics of sludge [163]. Currently, aluminum and iron are still not regarded as heavy metals, but a certain maximum limit for effluent or sludge is set by regulations. According to the Environmental Protection Agency (EPA 820-R-11-003), the maximum allowable concentrations of aluminum and iron in drinking water treatment sludge are 1 and 5 mg/L, respectively. As stated previously, the maximum allowable concentrations of aluminum and iron in wastewater sludge are still not yet regulated in many countries. 

In accordance with the mentioned characteristics, the suspended solids in water or industrial effluent, especially from agriculture, fermentation process, and the food industry, can actually be considered highly biodegradable organic sludge comprising BOD and nutrients. Nevertheless, when aluminum- and iron-based coagulants are applied to flocculate suspended solids, the characteristics of the resultant sludge make it minimally biodegradable. Most of the generated sludge is treated in regular landfills, which is considered an unsafe method, given that aluminum and iron contents have the potential to be leached out during acid rain [25,33]. A regular landfill has a thin base layer and limited protocols for handling metal leakage. The utilization of biocoagulants/bioflocculants will eliminate the abundant amount of aluminum/iron in the resultant sludge from effluent treatment. The disposal of sludge produced using biocoagulants/bioflocculants into regular landfills will be safe because it contains limited metal; in some conditions, it can be converted into a soil conditioner or fertilizer [142,143,164]. The utilization of biocoagulants/bioflocculants will also open up alternative treatment methods, including anaerobic digestion, which could lead to renewable energy production [33,58,165,166]. 

### 7.4. Operational Costs

The cost of alum utilization for drinking water treatment per m^3^ of treated water varies according to different sources, i.e., USD 0.05, as stated by Keeley et al. [167]; USD 0.10 as stated by Hamawand et al. [58]; and up to USD 1.50, as stated by Philip [168]. For wastewater treatment, the cost of using chemical coagulants/flocculants per m^3^ also varies, i.e., USD 0.10 when using alum and USD 0.15 when using ferric chloride to treat meat processing effluent [58], USD 19 to treat palm oil mill effluent by using alum [122], and up to USD 1.80 when using PAC for leachate treatment [169]. The costs of applying biocoagulants/bioflocculants to the treatment of wastewater and drinking water are listed in Table 4. 

The comparison of the costs of the chemical coagulants in Table 4 shows that the price of applying biocoagulants/bioflocculants is actually competitive with that of applying the conventional chemical ones commonly used. The application of some biocoagulants/bioflocculants shows a higher price compared with the conventional treatment using alum or ferric chloride, but the presented data indicate only the comparison of material utilization. A further comparison that might change the perspective of treatment is of the cost of the resultant sludge handling. As stated by Philip [168], the cost of drinking water sludge handling by landfilling reaches up to USD 1.50 per m^3^. The existence of chemicals, aluminum, and iron in sludge makes it difficult for it to be treated using biological treatment, such as anaerobic digestion, which is considered cheaper compared with landfilling, thus increasing the operational cost. The high bioavailability of aluminum and iron in sludge induces a toxic effect on the microorganisms in anaerobic digestion, resulting in low digestion performance [58]. The utilization of biocoagulants/bioflocculants reduces the existence of aluminum and iron in sludge and thus increases the biodegradability of the produced sludge to be treated using biological treatment. As mentioned, the operational cost of anaerobic digestion is slightly cheaper than that of landfilling or even sophisticated technologies, including electrochemical processes [24,33,58,174]. 

## 8. Limitations and Future Challenges

Despite all the mentioned advantages, the utilization of biocoagulants/bioflocculants still faces some limitations. The limitations and challenges, as summarized in Figure 6, are foreseen if biocoagulants/bioflocculants are to be applied in water or wastewater treatment.

### 8.1. Limitations of Current Studies

Unlike commercial coagulants/flocculants that are readily available to be purchased, the extraction of active compounds from biocoagulants/bioflocculants needs to go through a few stages before they can be used in the treatment. For plant-based coagulants, many processes are involved in the extraction/purification of active compounds for the coagulation–flocculation process. The extraction of plant-based biocoagulants/bioflocculants depends on the characteristics of the plants but is likely to be the same, including crushing, sieving, dialysis, washing, delipidation, the removal of insoluble matter, filtration, and ion exchange [175,176]. For animal-based biocoagulants/bioflocculants, the extraction processes include washing, drying, grinding, demineralization, deproteination, and deacetylation [177,178]. For microorganism-based biocoagulants/bioflocculants, the extraction processes include cultivation, centrifugation, the extraction of exopolymeric substances (EPS) by using saline solution or EDTA, filtration, and lyophilization [179,180,181]. These sequential processes need to be conducted carefully to retrieve the pure extracts of active compounds that could perform as biocoagulants. These long-winded processes need to be simplified to make the utilization of biocoagulants/bioflocculants applicable and readily available [182]. Further research should explore how to convert active compounds into powder form for easy handling and storage and preserve them for long-lasting characteristics.

In contrast to the use of iron and aluminum, which are abundant in the Earth, the application of biocoagulants/bioflocculants requires paying attention to the availability of the raw materials. The abundance of raw materials as biocoagulants/bioflocculants will make the application of this technology become realistic [134] and compatible with conventional coagulants and thus widen its application to a larger scale. Many researchers have suggested the utilization of local resources as raw materials for biocoagulants/bioflocculants [124,134,183]. Native/local resources are usually abundant, easy to be found, and renewable. The resources used as biocoagulants/bioflocculants are also suggested not to compete with other needs, such as basic foods, and to be sustainable and prevent scarcity in long-term application.

The different characteristics of water will affect the working conditions of the coagulants and flocculants and thus their performance and the efficiency of coagulation and flocculation processes [3]. Even for metal-based coagulants, which are already established, some working conditions need to be maintained or altered to achieve the optimum removal of pollutants. The application of biocoagulants/bioflocculants has been proven good for high-turbidity water, but limited efficiency has been achieved for low-turbidity water [45]. Despite the turbidity, the initial pH, total organic content, and electrostatic condition of the water and wastewater will have direct impacts on coagulation and flocculation processes using biocoagulants/bioflocculants [3,4].

The common coagulation and flocculation processes can also reduce the number of bacteria in effluent [123,184]. These processes provide advantages in reducing the load of disinfection units and help in complying to the stringent regulation standards. In view of biocoagulation/bioflocculation, limited research has been conducted [185]. The utilization of bacterium-based coagulants/flocculants also requires increasing the number of bacteria in the effluent, which could become a limitation of this technology. Consequently, the utilization of bacterium-based coagulants/flocculants might increase the load of disinfection units and could even make effluent fail to meet the regulation standards on the maximum number of bacteria. However, the extraction of active compounds expressed extracellularly by microorganisms and transforming them into powder form could prevent introducing additional bacteria to the treatment system, especially for drinking water treatment, which requires a disinfection process before water can be supplied for human consumption.

On the basis of all of the limitations, the application of biocoagulants/bioflocculants on a large/industrial scale will be challenging and remain at the research stage. The industrial application of this technology will cause challenges in terms of the massive production of biocoagulants/bioflocculants, the continuous supply of biocoagulants/bioflocculants, the abundant amount of raw materials, and the adjustment of all treatment equipment to provide optimum conditions for the processes of biocoagulation/bioflocculation. Further work needs to be conducted before the application of biocoagulants/bioflocculants can be compared with chemical coagulants.

### 8.2. Challenges for Future Studies

All the mentioned limitations of the application of biocoagulants/bioflocculants create challenges for researchers to further study these matters critically. Providing a simple and reliable method for extracting biocoagulant/bioflocculant compounds from raw materials could be the solution to overcoming the complex extraction of biocoagulants/bioflocculants. Many pathways of research into cutting the extraction processes can be carried out, considering that the processes depend on raw materials. The characterization of local abundant resources and their potential to be used as biocoagulants/bioflocculants for water and wastewater treatment is important to provide several options in the utilization of resources. A feasibility study and economic analysis on the potential of some waste, mostly in the agriculture and fisheries sectors, to be used as biocoagulants/bioflocculants will also be valuable in providing an abundant amount of less-wanted raw materials. The characterization of waste will provide options not only for reusing waste but also for contributing to the waste reduction effort. 

Determining the optimum conditions for coagulation and flocculation processes using biocoagulants/bioflocculants will be relevant to conduct in the future. An analysis of the suitability of using a certain type of biocoagulant/bioflocculant to treat a certain characteristic of water and wastewater will be useful in minimizing trial-and-error processes, given the many characteristics of wastewater. Thorough analysis also needs to be conducted on the utilization of bacterium-based coagulants/flocculants and its impact on the number of bacteria in effluent. Research on this topic is currently limited. Two researchers have reported the same result in relation to the use of plant-based coagulants and their effectiveness in removing coliforms by up to 100% [114,186]. Further assessment, especially on the use of bacterium-based coagulants/flocculants, is needed. 

In terms of flocculant production by microorganisms, current research is mostly conducted at the laboratory scale, while the mass production of flocculants is still beyond expectations [187,188]. The mass production of microorganism-based bioflocculant requires abundant and a continuous supply of cultivation medium [189]. Currently, the cultivation of flocculant-producing microorganisms still utilizes factory-made medium (conventional chemicals) as nutrients [187,188]. It is suggested that the mass production of microorganism-based bioflocculants should consider the utilization of waste products [190,191] or local resources as a growing medium [190,192]. Local resources are considered to be abundant and easy to acquire, while waste products are considered to be unwanted. The utilization of these materials may reduce production costs, creating sustainable chain-linked green treatment, while also unlocking potential in mass production for real-scale application. 

Lastly, the greatest challenge for this technology is its introduction at a larger scale for industrial applications [193,194]. The green technology, in industry, is currently growing and capturing considerable attention due to the toxic impacts of conventional coagulation. Hence, research on scaling up the laboratory findings and performing the continuous treatment of water or wastewater will be genuinely motivating not only for academic-background readers but also for professional- and industrial-background readers.

## 9. Conclusions

Several studies related to the application of biocoagulants/bioflocculants are currently being conducted at the laboratory scale. However, research on the application of this technology on a pilot or industrial scale is limited and still scattered. The utilization of biocoagulants/bioflocculants is a promising technology for application for treating water and wastewater because it is environmentally friendly and publicly accepted and has reliable performance. Nonetheless, several factors need to be considered as limitations in applying this technology due to the complex extraction process, the limited availability of the raw materials, the various characteristics of water and wastewater to be treated, a consideration of the increasing bacterial count in the effluent, and the application of this technology at an industrial scale. The limitations of applying this technology open up new challenges for future study. Opportunities include research on simplifying the extraction methods, characterizing the potential of local resources to be used as biocoagulants/bioflocculants, determining the optimum working conditions, analyzing the capability in removing bacteria, and scaling up the laboratory research to an industrial scale. In consideration of all the advantages, the development of biocoagulants/bioflocculants for water and wastewater treatment processes has good prospects as a green technology for treating effluent and yielding value-added products, such as fertilizers/soil conditioners, in the future. 

## Figures and Tables

**Figure 1 ijerph-17-09312-f001:**
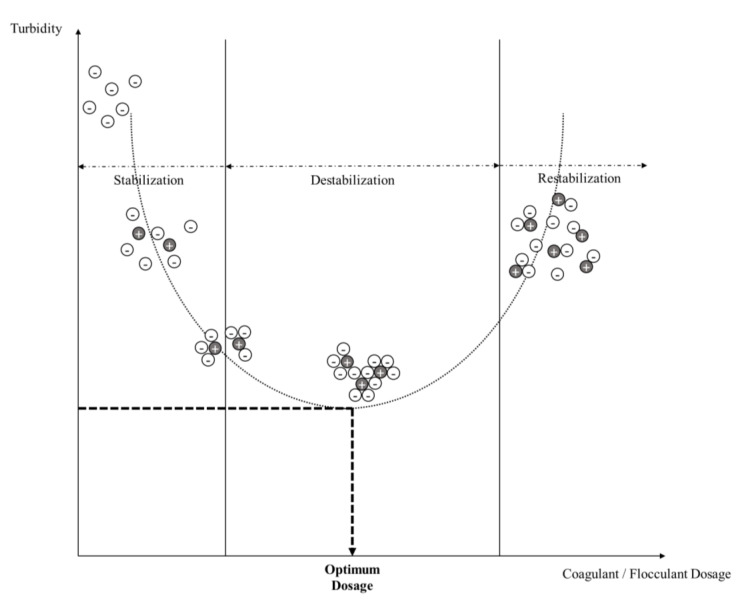
Phases through the coagulation–flocculation process.

**Figure 2 ijerph-17-09312-f002:**
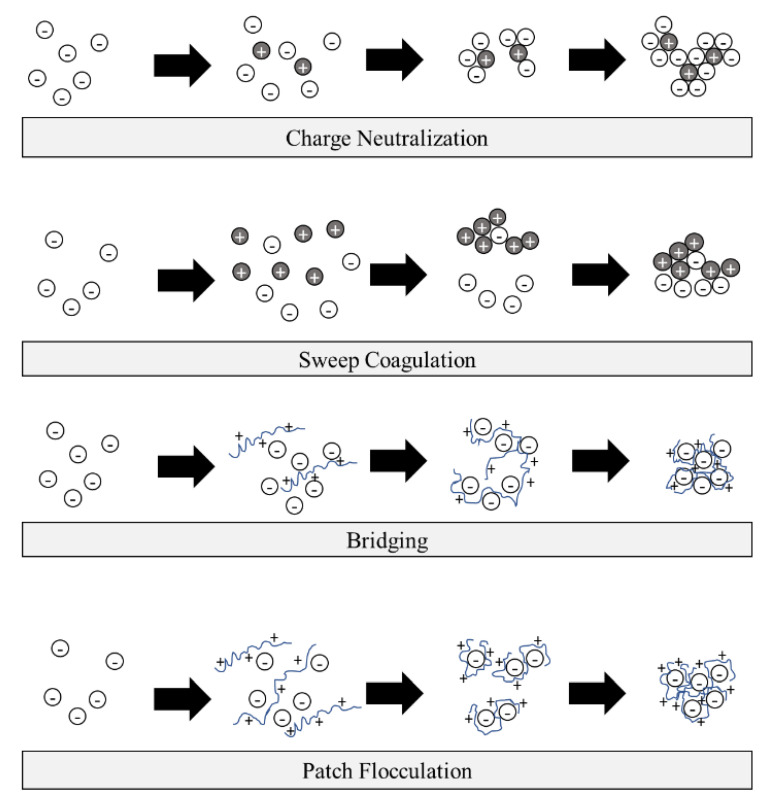
Mechanisms of coagulation and flocculation.

**Figure 3 ijerph-17-09312-f003:**
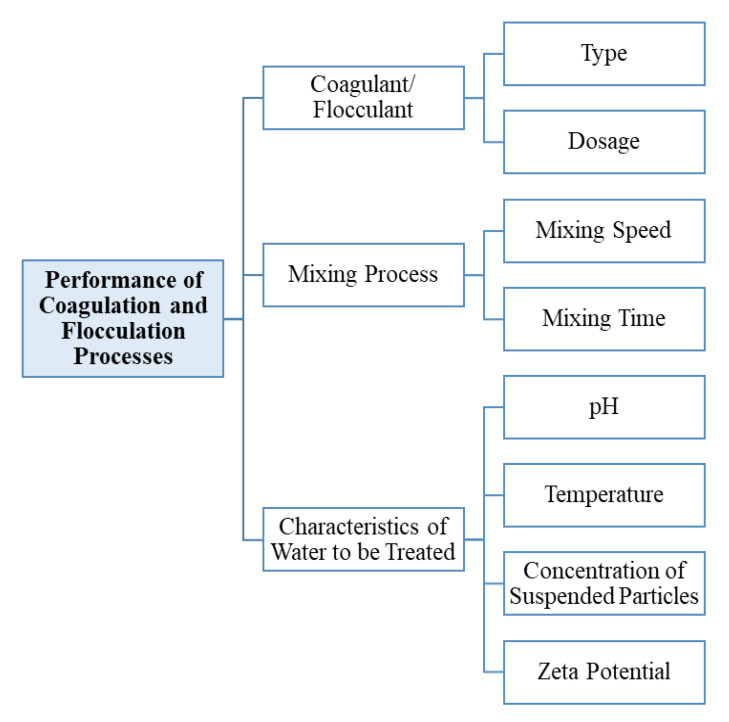
Factors affecting coagulation and flocculation processes.

**Figure 4 ijerph-17-09312-f004:**
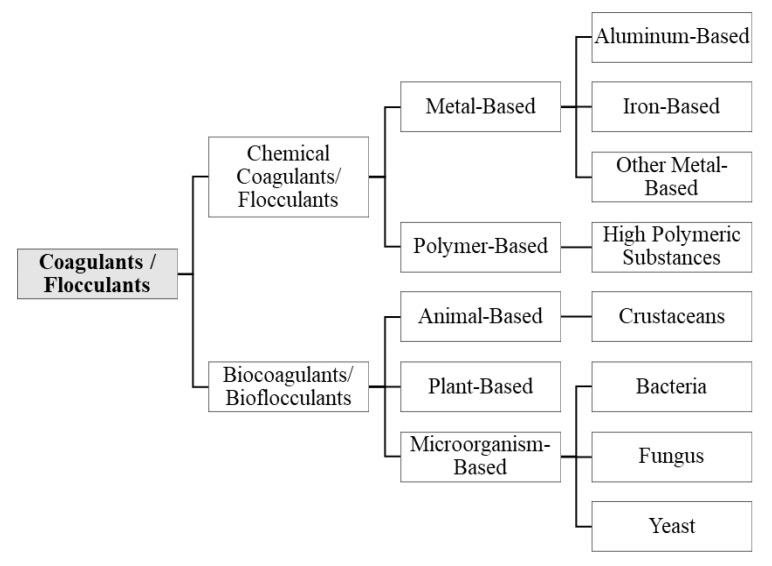
Classification of coagulants/flocculants.

**Figure 5 ijerph-17-09312-f005:**
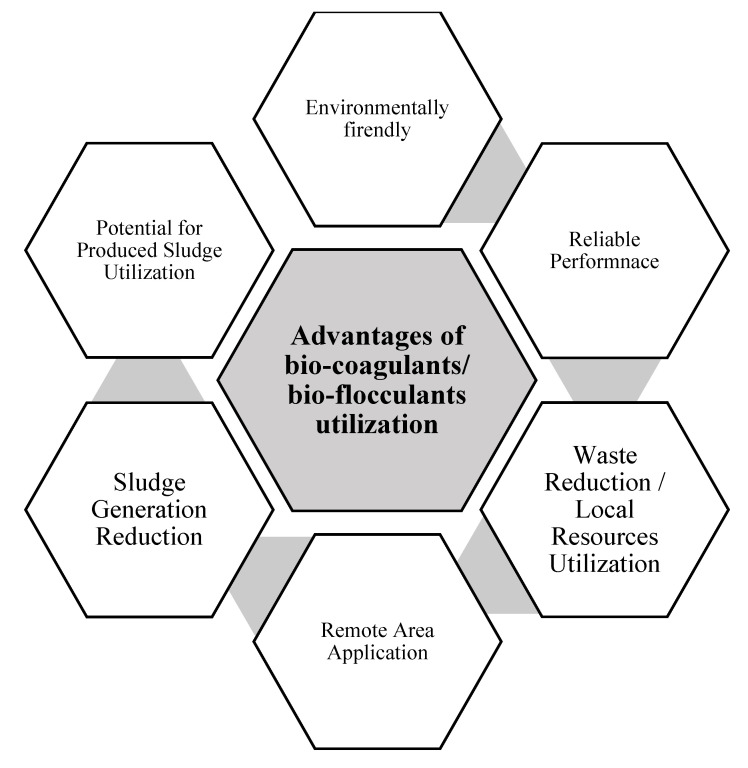
Advantages of biocoagulant/bioflocculant utilization in drinking water/wastewater treatment.

**Figure 6 ijerph-17-09312-f006:**
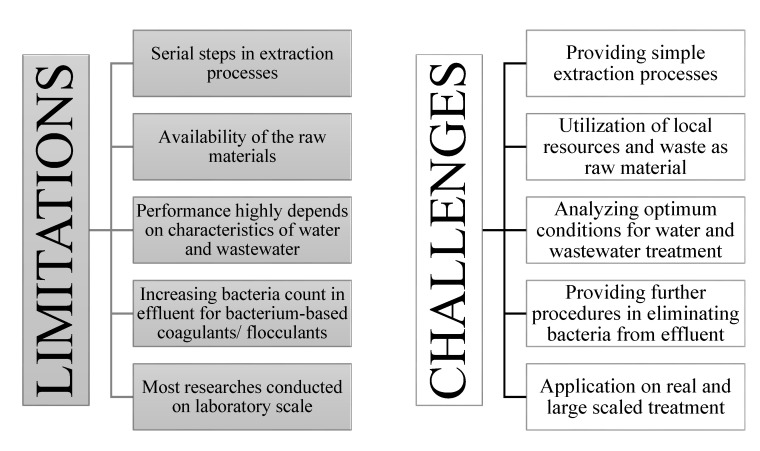
Limitations and challenges of biocoagulant/bioflocculant application.

**Table 1 ijerph-17-09312-t001:** Characterization of chemical contents in biocoagulants/bioflocculants.

No	Type	Species	Chemical Compounds/Functional Groups	Source
1	Animal-based	Shellfish	Chitin and polysaccharides	[33]
2	Animal-based	Shrimp shell	Chitosan and carboxy methyl cellulose	[95]
3	Animal-based	Periwinkle shell	Alcohol, phenol, secondary amide group, amine group, alkyne group, and polysaccharides	[96]
4	Animal-based	Crab shell	Chitosan	[97]
5	Microorganism-based (bacteria)	*Bacillus agaradhaerens* C9	Carboxyl, hydroxyl, amino, and glycoprotein groups	[98]
6	Microorganism-based (bacteria)	*Bacillus mucilaginosus*	Uronic acid, neutral sugar, amino sugar, carboxyl group, and hydroxyl group	[99]
7	Microorganism-based (bacteria)	*Bacillus salmalaya* 139SI-7	Carboxyl group, hydroxyl group, amino group, polysaccharides, and proteins	[100]
8	Microorganism-based (bacteria)	*Paenibacillus polymyxa*	Polysaccharides and proteins	[101]
9	Microorganism-based (bacteria)	*Bacillus licheniformis* strain W7	Polysaccharides, protein, hydroxyl group, carboxyl group, and amino group	[51]
10	Microorganism-based (bacteria)	*Bacillus velezensis*	Xylose and glucose	[102]
11	Plant-based	Rice starch	Cellulose, lignin, aldehydes, ketones, esters, and carboxylic acids	[103]
12	Plant-based	*Lens culinaris*	Hydroxyl and carboxyl groups	[104]
13	Plant-based	Cassava	Amino acids, carboxyl group, and hydroxyl group	[105]
14	Plant-based	*Dillenia indica*	Polysaccharides	[94]
15	Plant-based	Potato starch	Branched-structure polymers	[106]
16	Plant-based	*Moringa oleifera* seed	Cationic protein, starch, glucose, fatty acids, and phenolic compounds	[107]
17	Plant-based	*Moringa oleifera* seed	Alcoholic compound, polysaccharides, and amides	[108]
18	Plant-based	*Moringa oleifera* seed	Amines, carboxylate groups, and alcoholic compounds	[109]

**Table 2 ijerph-17-09312-t002:** Potential of sludge recovery/utilization from wastewater.

No	Sludge Recovery/Utilization	Type of Wastewater	Summary of Findings	Country	Source
1	Algae biomass	Aquaculture	Chlorella sp. was recovered from aquaculture wastewater. The recovery process needs to be conducted with biodegradable coagulants in order to use the sludge further. Chitosan can perform 80% algae biomass recovery after coagulation–flocculation and sedimentation processes. Recovered biomass can be utilized as feed inside the biofloc cultivation pond system.	Malaysia	[39]
2	Algae biomass	Aquaculture	Derivates of *Moringa oleifera* showed potential to be used as bioflocculant to recover Chlorella sp. from aquaculture effluent. Biomass recovery efficiency using *Moringa oleifera* was significantly higher as compared to commercial chemical coagulant.	Malaysia	[145]
3	Nutrient into fertilizer	Domestic	Sludge of domestic wastewater contains high concentrations of nitrogen and phosphorus, which are essential nutrients for plant growth. Recovered sludge from domestic wastewater even contains trace elements, which might boost plant growth. Utilization of sludge from domestic wastewater needs to be accompanied by the utilization of biodegradable coagulants.	Eswatini	[143]
4	Nutrient into fertilizer	Domestic	Recovered domestic sewage sludge characterized as organic-rich solid may be very useful to provide readily bioavailable macro- and micro-components for plant growth. The result showed that the organic content in sludge can act as a soil conditioner.	Poland	[142]
5	Nutrients into fertilizer	Aquaculture	Recovered sludge from aquaculture effluent was used as biofertilizer for hot pepper plants. Positive effect on the plant height was obtained and considered to be comparable to that with the utilization of commercial fertilizer.	Nigeria	[146]
6	Nutrients into fertilizer	Domestic	Domestic wastewater contains high amounts of nutrients, which can be recovered as fertilizer. Recovered sludge as fertilizer has a characteristic of slow-release activity, which is useful for providing nutrients over a longer period of time.	Ohio	[137]

**Table 3 ijerph-17-09312-t003:** Performance of biocoagulants/bioflocculants in treating drinking water and wastewater.

No.	Name	Type	Function	Treated Water	Summary	Country	Source
1	*Achatinoidea* shell	Animal-based	Biocoagulant	Paint industry wastewater	*Achatinoidea* shell could reduce total dissolved solid (TDS) by up to 13% for 35 min of settling time with a dosage of 4 g/L at pH 7.9. Optimum performance of 99.22% was obtained at pH 4, 4 g/L dosage, and 45 °C.	Texas	[153]
2	Crab shell	Animal-based	Biocoagulant	Lake water	The crab shell could aid alum as a biocoagulant to enhance turbidity removal (97%) with 0.2 mg/L dosage after 45 min of settling time. Crab shell could be used as a natural aid coagulant for drinking water treatment with the lowest risks of organic release.	Algeria	[97]
3	Crab shells	Animal-based	Biocoagulant	Drinking water	Combining crab shell as biocoagulant and alum could reduce turbidity of low-, medium-, and high-turbidity water by up to 74.8%, 96.7%, and 98.2%, respectively. This removal was higher than that using only alum as coagulant. This biocoagulant could reduce the alum dose by up to 75%, and the sludge by-product is readily biodegradable. The optimum pH and biocoagulant dose for removing turbidity were 7 and 1.5 mg/L, respectively.	India	[112]
4	Periwinkle shell	Animal-based	Biocoagulant	Petroleum wastewater	Varying the dosage of periwinkle shell and pH had a significant effect on the coagulation–flocculation efficiency. The optimum conditions were pH 4 and a 100 mg/L periwinkle shell dosage. The removal of particles was up to 83.57%.	Texas	[96]
5	Shrimp shells	Animal-based	Biocoagulant	Wastewater containing oil	The chitosan from shrimp shell as a biocoagulant could reduce oil by up to 96.35% at pH 4 over 60 min of contact time. The removal of oil by using chitosan was increased after adding carboxy methyl cellulose (CMC), with percentage efficiency of 99% at (90% chitosan and 10% CMC) with 30–60 min of contact time.	Egypt	[95]
6	Snail shell	Animal-based	Biocoagulant	Wastewater containing dye	The snail shell alone as biocoagulant could reduce malachite green (MG) dye by up to 60% with a dosage of 100 mg/L. The combination of snail shell and alum could enhance the removal of MG dye. The optimum pH for MG dye removal was found to range between 4 and 5. The optimum flocculation time was 30 min with an alum–snail shell dosage of 20–100 mg/L. The sludge produced from the alum–snail shell combination had better settling characteristics than the sludge obtained from the use of snail shell alone.	Nigeria	[154]
7	Alginate	Microorganism-based (algae)	Biocoagulant	Drinking water	Algal alginate has a high polysaccharide content that could perform as a biocoagulant. Alginate removed up to 98% of suspended solids from high-turbidity water. A low dosage of the coagulant (as low as 0.02 mg/L) still achieved high turbidity removal.	Turkey	[155]
8	*Achromobacter xylosoxidans* strain TERI L1	Microorganism-based (bacteria)	Bioflocculant	Wastewater containing heavy metals	*Achromobacter xylosoxidans* strain TERI L1 could produce exopolysaccharide as a bioflocculant. The bioflocculant contained 75% total sugar, with 72.9% neutral sugar and 11.5% protein. *Achromobacter xylosoxidans* strain TERI L1 could flocculate Zn, Pb, Ni, Cd, and Cu by up to 90%.	India	[98]
9	*Bacillus agaradhaerens* C9	Microorganism-based (bacteria)	Bioflocculant	Wastewater containing microalgae	A bioflocculant was extracted from *Bacillus agaradhaerens* C9 and contained 65.42% polysaccharides, 4.70% proteins, and 1.65% nucleic acids. The optimum conditions for producing bioflocculant from *Bacillus agaradhaerens* C9 were 10 g/L of glucose, 10 g/L of yeast extract, and an initial pH of 10.2. The flocculation rate for kaolin suspension was 95.29%, with optimum dosage, pH, and temperature of 1.5 mg/L, 6.53, and 29 °C, respectively. The bioflocculant had the potential to treat alkaline wastewater.	China	[156]
10	*Bacillus licheniformis* strain W7	Microorganism-based (bacteria)	Bioflocculant	Synthetic wastewater containing kaolin and river water	A bioflocculant (MBF-W7) was produced using *Bacillus licheniformis* strain W7. The optimum conditions for flocculant production were a 5% (*v/v*) inoculum size with maltose and NH_4_NO_3_ as carbon and nitrogen sources. The pH and cultivation time were 6 and 72 h, respectively. The flocculation rate for kaolin clay suspension was 85.8%, observed at pH 3, and MBF-W7 of 0.2 mg/mL. MBF-W7 could remove turbidity and chemical oxygen demand (COD) by up to 86.9% and 75.3%, respectively, in Tyume River.	South Africa	[51]
11	*Bacillus mucilaginosus*	Microorganism-based (bacteria)	Bioflocculant	Starch wastewater	A bioflocculant (MBFA9) was produced from *Bacillus mucilaginosus*. The major component was a polysaccharide that contained uronic acid (19.1%), neutral sugar (47.4%), and amino sugar (2.7%). The flocculation rate for kaolin suspension was 99.6% with a 0.1 mL/L MBFA9 dosage. MBFA9 could reduce total suspended solid (TSS) and COD by up to 85.5% and 68.5%, respectively.	Singapore	[99]
12	*Bacillus salmalaya* 139SI-7	Microorganism-based (bacteria)	Bioflocculant	Organic-rich wastewater	A bioflocculant (QZ-7) was synthesized using *Bacillus salmalaya* strain 139SI with flocculation activity of 83.3%. The optimum temperature, pH, and incubation time conditions for flocculant production were 35.5 °C, 7, and 72 h, respectively, with inoculum size of 5% (*v/v*), sucrose as carbon source, and yeast extract as nitrogen source. Bioflocculant QZ-7 could remove COD and BOD by 93% and 92.4%, respectively.	Malaysia	[100]
13	*Bacillus velezensis*	Microorganism-based (bacteria)	Bioflocculant	Lake water	This study investigated the effects of incubation time and temperature on the production of bioflocculants by using *Bacillus velezensis* grown in sago mill effluent (SME) and palm oil mill effluent (POME) as a fermentation feedstock. The highest bioflocculant yield (2.03 g/L) at a temperature of 40 °C was achieved in POME medium. The bioflocculant produced from a fermented SME medium (BioF-SME) showed the highest activity. Bioflocculants from POME and SME had performance comparable with alum’s in removing color and turbidity from lake water.	Malaysia	[102]
14	*Chromobacterium violaceum* and *Citrobacter koseri*	Microorganism-based (bacteria)	Bioflocculant	Tapioca wastewater	*Chromobacterium violaceum* and *Citrobacter koseri* were isolated from tapioca wastewater and had high flocculation activities of 68.92% and 71.38%, respectively. The optimum pH and temperature for *Chromobacterium violaceum* and *Citrobacter koseri* were 2–4 and 6–8 and 40 °C and 30 °C, respectively.	Indonesia	[157]
15	*Paenibacillus polymyxa*	Microorganism-based (bacteria)	Bioflocculant	Formaldehyde wastewater	A novel bioflocculant-producing bacterium (MBF-79) was isolated from formaldehyde wastewater sludge. The optimum inoculum size, pH, and formaldehyde concentration for bioflocculant production were 7.0%, 6, and 350 mg/L, respectively. The major components of MBF-79 were polysaccharide (71.2%) and protein (27.9%). The optimum MBF-79, pH, contact time for the removal of arsenate and arsenite by using MBF-79 were 120 mg/L, 7, and 60 min, respectively, with removal efficiencies of 98.9% and 84.6%, respectively.	China	[101]
16	*Aspergillus niger*	Microorganism-based (fungi)	Bioflocculant	Aquaculture wastewater	*Aspergillus niger* was applied to flocculate microalgae from aquaculture wastewater. More than 90% harvesting efficiency was obtained at pH 3.0 to 9.0 and a mixing rate of 100–150 rpm.	Malaysia	[152]
17	*Aspergillus niger*	Microorganism-based (fungi)	Bioflocculant	Potato starch wastewater	Two milliliters of the bioflocculant produced using *A. niger* was able to remove up to 91.15% of COD and 60.22% of turbidity within 20 min of treatment. Compared with the conventional coagulants (alum- and iron-based), this bioflocculant showed nearly identical performance with a lower material cost and a smaller yield of sludge.	Hong Kong	[151]
18	*Penicillium sp.* and *Trichoderma sp.*	Microorganism-based (fungi)	Biocoagulant	Domestic wastewater	Suspension of fungal spores was proven to reduce 84% (relative to alum efficiency) of turbidity from sewage at pH 7.8 with 60 min of treatment.	Iraq	[150]
19	*Abelmoschus esculentus*	Plant-based	Biocoagulant	Industrial textile wastewater	*Abelmoschus esculentus* as biocoagulant is more efficient for treating textile wastewater than chloride ferric. *Abelmoschus esculentus* can remove turbidity, COD, and color by up to 97.25%, 85.69%, and 93.57%, respectively, with optimum pH and concentration of biocoagulant of 6 and 3.2 mg/L, respectively.	Brazil	[57]
20	Dragon fruit foliage	Plant-based	Biocoagulant	Concentrated latex wastewater	Dragon fruit foliage as biocoagulant could reduce COD, SS, and turbidity from latex effluent by up to 94.7%, 88.9%, and 99.7%, respectively, at pH 10. The biocoagulant dosage range of 200–800 mg/L showed consistent removal of pollutants. The removal percentage for pollutants using ferric sulfate was higher than that using dragon fruit foliage.	Malaysia	[148]
21	*Moringa oleifera*	Plant-based	Biocoagulant	Synthetic turbid wastewater	Raw *Moringa oleifera* seed contains high amounts of oil, which can reduce the potential for coagulation activity. Oil extraction significantly increased the coagulation activity of *Moringa oleifera* seed. The utilization of this biocoagulant showed 82.43% oil and grease removal from water.	Brazil	[107]
22	*Moringa oleifera*	Plant-based	Biocoagulant	Hospital wastewater	*Moringa oleifera* extract contains dimeric protein. Utilization of this biocoagulant showed 65% removal of turbidity, 38% of COD, and up to 90% removal of *Pseudomonas aeruginosa.*	Benin	[108]
23	*Moringa oleifera*	Plant-based	Biocoagulant	Drinking water	Integrating seed powder of *Moringa oleifera* into solar water disinfection could reduce turbidity by up to 85% in 24 h and remove *Escherichia coli* in 6 h.	Ireland	[158]
24	*Moringa oleifera*	Plant-based	Biocoagulant	Freshwater containing microalgae	*Moringa oleifera* (MO) seed derivatives were used to harvest suspended microalgae, *Chlorella* sp. Flocculation efficiency of more than 95% was achieved with 20 min sedimentation. MO derivatives had better performance compared with aluminum sulfate at a low dosage of 10 mg L^−1^ and normal pH (6.9–7.5).	Malaysia	[145]
25	*Ocimum basilicum* L.	Plant-based	Biocoagulant	Leachate	*Ocimum basilicum* L. has potential as biocoagulant for leachate pretreatment. Combining *O. basilicum* and alum as coagulant could remove COD and color by up to 64.4% and 77.8%, respectively, with optimum conditions of 15 min of settling time, pH of 7, and alum/*O. basilicum* ratio of 1:1. Integrating biocoagulant of *O. basilicum* and ozonation could increase the percentage removal of COD and color by up to 92% and 87%, respectively.	Iran	[159]
26	Rice starch	Plant-based	Biocoagulant	Palm oil mill effluent (POME)	The floc shaped by rice starch was more stable than alum. The rice starch as biocoagulant could reduce TSS from POME by up to 84.1% with optimum conditions of dosage, pH, settling time, and slow stirring speed of 2 g/L, pH 3, 5 min, and 10 rpm, respectively. Combining rice starch (0.55 g/L) and alum (0.2 g/L) could increase the removal of TSS from POME by up to 88.4%.	Malaysia	[103]

**Table 4 ijerph-17-09312-t004:** Estimated costs of biocoagulant/bioflocculant application.

No.	Treated Water	Type	Estimated Cost (USD per m^3^)	Source
1	Drinking Water	Chitosan	0.0025	[170]
2	Drinking Water	*Moringa oleifera*	0.75	[171]
3	Drinking Water	*Parkinsonia aculeata*	2	[172]
4	Wastewater	*Moringa oleifera*	2	[35]
5	Wastewater	*Azadirachta indica*	6.8	[121]
6	Wastewater	*Moringa oleifera*	5.2	[121]
7	Wastewater (Domestic)	Chitosan	0.015	[58]
8	Wastewater (Leachate)	*Tamarindus indica*	19.50	[169]
9	Wastewater (Palm Oil Mill Effluent)	Water-soluble chitosan from mushrooms	1.25	[122]
10	Wastewater (Palm Oil Mill Effluent)	Acid-soluble chitosan from mushrooms	1	[122]
11	Wastewater (Paper Mill Effluent)	Starch	1–2.85	[173]

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
