# Peer review of "Challenges and Opportunities of Biocoagulant/Bioflocculant Application for Drinking Water and Wastewater Treatment and Its Potential for Sludge Recovery"

_ijerph, 2020, doi:10.3390/ijerph17249312_

Round 1
Reviewer 1 Report
It is interesting reading this high-quality review manuscript on bioflocculant research. The authors have done a great done and they have contributed to the body of knowledge in this field and I believe they are experts in the field because of the state of the art of relevant literature presented in the paper. The English use in the manuscript is fine but still, needs minor editing. However, I noted some corrections that could improve the quality of the manuscript further.
- It would be appropriate for the authors to have a section in the manuscript that would discuss the use of cost-effective substrates for bioflocculant production because of the cost implication of the conventional chemicals used as a nutrient in the media.
- There should be a section that will highlight the various factors affecting bioflocculant flocculating activity.
- Revise the statement in lines 22-23. Too many “and”
- Line 29, change “to” to “in”
- Line 66, change “applying” to “involving”
- Line 92, change “are” to “were”
- There are several scientific statements that needed to be backed up with appropriate references. Lines 45, 48, 50, 52, 54, 60, 62, 68, 71, 72, 76, 80, 92, 101, 105, 108, 118, 140, 141, 148, 243.
Author Response
Response to Reviewer 1 comments are provided as attachment. Kindly refer to the attached file please.

Reviewer 2 Report
The intent of this study is good. The paper is fairly well-written. Study results may add to the existing knowledge. However, the following comments may further enhance the readability of this manuscript:
- Check to be sure that all acronym names have been defined when first appear.
- In Line 370, Are there “four” major advantages instead of “six” illustrated in Figure 5?
- Heading for Section 7 is not clear and appropriate.
- In Line 538, is the cost to treat palm oil mill USD 19 or UDS 1.25 (as mentioned in Table 4)?
- Table results have taken up too much space. Substantial size reduction is necessary to enhance readability.
- Review results may be segregated into “drinking water” and “waste water” to increase clarity.
- Similarly, limitations and challenges may be segregated to enhance review presentation clarity.
- Some English language errors have been detected (Lines 259-260, 615).
Author Response
Response to Reviewer 2 comments are provided as attachment. Kindly refer to the attached file please.

Reviewer 3 Report
Finding new alternative coagulants/flocculants with lower footprints and toxicities is a very important aspect of water purification. However, authors enumerate, systematically, well-known concepts about biosourced coagulants and flocculants. The beginning of the manuscript is poor of new elements and the literature review is seriously insufficient (see proposed references below). However, novels aspects are discussed in sections 5, 6 and 7. If sections 2, 3 and 4 are compressed and improved considering the state of the art, this manuscript could be accepted after major revision.
Line 54: negatively charged polymers are also largely used, notably as high molecular weight flocculant.
Line 74: I feel the literature review about Al impact on health and Alzheimer’s disease could be improved - some papers mentioned the risk is relatively low due to the very low Al concentration in treated water.
Line 82: The authors missed two important literature review on the subject:
1) Organic polyelectrolytes in water treatment (2007)
2)Understanding the roles and characterizing the intrinsic properties of synthetic vs. natural polymers to improve clarification through interparticle Bridging: A review (2020)
I think the manuscript could be enhanced - clear comments must be provided on those two review papers e.g., attachment mechanisms, molecular weight and polymer configuration (linear or ramified).
--
Section 2 is common knowledge and could be drastically reduced to more focus on the novel aspects proposed in the manuscript.
--
The impact (toxicity) of acrylamide-based flocculants, and acrylamide itself, is not covered in section 3, while it is an important aspect and draw of such flocculant (one of the most used globally). Please see:
1) Substituting polyacrylamide with an activated starch polymer during ballasted flocculation (2019)
--
Section 4 is good and well-designed. However, potato starch is not covered, while it becoming more popular in Europe and North America. If needed:
1) Dual starch–polyacrylamide polymer system for improved flocculation (2017)
Section 4.3.: again, the authors forgot to mention one important review paper in the field of the soluble polymers used as flocculants (from Gregory and Barany):
1) Adsorption and flocculation by polymers and polymer mixtures
--
Sections 5, 6 and 7 are also well-designed.
Author Response
Response to Reviewer 3 comments are provided as attachment. Please see the attached file.

Round 2
Reviewer 3 Report
N.A.